# Sex and gender differences in presentation, treatment and outcomes in acute coronary syndrome, a 10 year study from a multi-ethnic Asian population: The Malaysian National Cardiovascular Disease Database—Acute Coronary Syndrome (NCVD-ACS) registry

**Chuey Yan Lee** [1]*, **Kien Ting Liu**[2], **Hou Tee Lu**[1], **Rosli Mohd Ali**[3], **Alan Yean Yip Fong** [4,5], **Wan Azman Wan Ahmad**[6]

**1** Department of Cardiology, Sultanah Aminah Hospital, Ministry of Health, Johor Bahru, Johor, Malaysia, **2** National Heart Association of Malaysia, Kuala Lumpur, Malaysia, **3** Cardiac Vascular Sentral Kuala Lumpur, Kuala Lumpur, Malaysia, **4** Department of Cardiology, Sarawak Heart Centre, Ministry of Health, Kota Samarahan, Sarawak, Malaysia, **5** Clinical Research Centre, Sarawak General Hospital, Institute for Clinical Research, National Institute of Health, Kuching, Sarawak, Malaysia, **6** University Malaya Medical Centre, Kuala Lumpur, Malaysia

* chueyyanlee@gmail.com

## Abstract

### Background

Sex and gender differences in acute coronary syndrome (ACS) have been well studied in the western population. However, limited studies have examined the trends of these differences in a multi-ethnic Asian population.

### Objectives

To study the trends in sex and gender differences in ACS using the Malaysian NCVD-ACS Registry.

### Methods

Data from 24 hospitals involving 35,232 ACS patients (79.44% men and 20.56% women) from 1st. Jan 2012 to 31st. Dec 2016 were analysed. Data were collected on demographic characteristics, coronary risk factors, anthropometrics, treatments and outcomes. Analyses were done for ACS as a whole and separately for ST-segment elevation myocardial infarction (STEMI), Non-STEMI and unstable angina. These were then compared to published data from March 2006 to February 2010 which included 13,591 ACS patients (75.8% men and 24.2% women).

**Data Availability Statement:** All relevant data are within the manuscript and its Supporting information files.

**Funding:** The National Cardiovascular Disease Database-Acute Coronary Syndrome (NCVD-ACS) Registry is funded by the Ministry of Health Malaysia (MOH), URL: https://www.moh.gov.my/ and the National Heart Association of Malaysia (NHAM), URL: https://www.malaysianheart.org/ The funders had no role in the study design, data collection and analysis, decision to publish or preparation of the manuscript.

**Competing interests:** The authors have declared no competing interests exist.

## Results

Women were older and more likely to have diabetes mellitus, hypertension, dyslipidemia, previous heart failure and renal failure than men. Women remained less likely to receive aspirin, beta-blocker, angiotensin-converting enzyme inhibitor (ACE-I) and statin. Women were less likely to undergo angiography and percutaneous coronary intervention (PCI) despite an overall increase. In the STEMI cohort, despite a marked increase in presentation with Killip class IV, women were less likely to received primary PCI or fibrinolysis and had longer median door-to-needle and door-to-balloon time compared to men, although these had improved. Women had higher unadjusted in-hospital, 30-Day and 1-year mortality rates compared to men for the STEMI and NSTEMI cohorts. After multivariate adjustments, 1-year mortality remained significantly higher for women with STEMI (adjusted OR: 1.31 (1.09–1.57), $p<0.003$) but were no longer significant for NSTEMI cohort.

## Conclusion

Women continued to have longer system delays, receive less aggressive pharmacotherapies and invasive treatments with poorer outcome. There is an urgent need for increased effort from all stakeholders if we are to narrow this gap.

## Introduction

Cardiovascular disease (CVD) remained the most common cause of death in women worldwide [1]. In Europe it is responsible for 49% of deaths in women and 40% of deaths in men [2, 3]. Recent studies have also reported significant increase in case fatality rates of acute coronary syndrome (ACS) in young women <55 years of age, while a decrease in mortality from coronary artery disease (CAD) in younger men [4–6]. In Asia-Pacific countries where rapid urbanisation and lifestyle changes are occurring, CVD is on the rise.

Over the past three decades there is growing evidence demonstrating differences between women and men in epidemiology, risk factors, clinical manifestations, diagnoses, treatment efficacies and outcomes of ACS [7–10]. These differences arise due to biological differences among women and men, called sex differences; and gender differences which are unique to human [11]. Sex differences are due to differences in gene expression from the sex chromosomes and subsequent differences in sexual hormones leading to differences in gene expression and function in the cardiovascular system. Whereas, gender differences arise from socio-cultural processes such as different behaviors of women and men, exposure to specific influences of the environment, different forms of nutrition, lifestyle, or stress, or attitudes towards treatments and prevention. Both these are equally important in CVD and as it is almost impossible to distinguish distinctly between the effects of sex and gender, we will discuss both of them together.

Many studies have shown women to be older and have more comorbidities [8, 9, 12–15]. Women were persistently under-represented in clinical trials and were also less likely to receive interventions such as coronary angiography, percutaneous coronary intervention (PCI) or coronary artery by-pass graft surgery (CABG) [8, 9, 15]. Some studies [8, 9, 16–19], but not all [13], had shown women with ACS had worse in-hospital and long-term prognoses than men. However, after adjustment for comorbidities and confounding factors, there was no difference in mortality between sexes [8, 20, 21].

These sex differences in ACS have been well-studied in registries and clinical trials in the developed western countries whereby men constitute the majority of the cohort. However, data looking at temporal changes over time is sparse from Asian countries, more so in a developing, multi-ethnic population. It may not be appropriate to apply the findings from a western population onto our local population.

We aim to study the temporal changes in the differences and similarities between women and men diagnosed with ACS using the NCVD-ACS registry. This registry is a joint effort between physicians and nurses in public, private and academic medical institutions. It is supported by the National Heart Association and the Ministry of Health of Malaysia [22].

## Methods

The Malaysian NCVD-ACS is an on-going, multi-center, observational prospective registry of patients presenting with acute coronary syndrome (ACS) i.e. ST-segment elevation myocardial infarction (STEMI), Non-ST-segment elevation myocardial infarction (NSTEMI) and unstable angina (UA). It was started in 2006 and was designed to evaluate the clinical presentation, management and clinical outcomes of patients, 18 years and above who presented with clinical features consistent with ACS accompanied by electrocardiographic and biochemical features. An overview of the Malaysian NCVD-ACS registry [22], methods and annual reports have been published elsewhere [23]. Full details of the methods of analysis on sex differences in ACS have been described in the first publication [12]. The current analysis is based on data collected from 1st. January 2012 through 31st. December 2016 from 19 public hospitals nationwide (10 tertiary hospitals with cardiology department/units and 9 tertiary and secondary hospitals without cardiologists), 3 academic teaching hospitals and 2 private hospitals (one of which is the National Heart Institute). Center participation was voluntary and all consecutive patients 18 years and above presenting with ACS were recruited. It comprised of 35,232 ACS patients, of which 79.44% were men and 20.56% were women. Of these 16,768 (47.59%) presented with STEMI, 9,209 (26.14%) presented with NSTEMI and 9,255 (26.27%) presented with UA. These were then compared to the data from March 2006 to February 2010 which included 13,591 ACS patients, of which 10,299 (75.8%) were men and 3,292 (24.2%) were women from 15 tertiary public hospitals (10 with Cardiology departments/units, 5 without Cardiologist), 1 academic teaching hospital and 1 private hospital (the National Heart Institute) [12].

Data were collected on demographic characteristics (age, sex and ethnicity), coronary risk factors (cigarette smoking, diabetes mellitus [DM], hypertension and dyslipidemia) and other comorbidities (previous myocardial infarction [MI], heart failure, or renal failure; cerebrovascular accident and body mass index). The vital signs at presentation, time-to-treatment (door-to-needle time and door-to-balloon time for STEMI cohort), in-hospital medical and invasive treatments, disease severity (culprit artery and number of diseased vessels) and in-hospital outcomes (all-cause mortality, hospitalisation days [coronary care unit and total days], and complications [bleeding rates] were also captured [12]. For the present cohort all-cause mortality outcome analysis were extended to Day-30 and 1-year. Mortality data were cross-checked with the National Registration Department. Analyses were done for ACS as a whole and for each stratum of ST-segment elevation myocardial infarction (STEMI), Non-STEMI (NSTEMI) and unstable angina (UA).

### Ethics approval

The NCVD-ACS registry is registered with the National Medical Research Register of Malaysia (ID: NMRR-07-38-164) and received ethical approval from the Medical Research and Ethics

Committee (MREC), Ministry of Health Malaysia in 2009. The Medical Review and Ethics Committee also waived the need for informed consent.

## Statistical analysis

Descriptive statistics and baseline variables are presented as numbers and percentages, mean ± SD, or median (interquartile ranges [IQR]). A Chi-square test was used to assess differences between categorical variables, an independent *t* test (parametric analysis) or Mann-Whitney *U* test (non-parametric analysis) was used to test differences between numerical variables. For multivariate analyses, simple binary and multiple logistic regressions were used to model the dichotomous outcomes of STEMI, NSTEMI and UA in-hospital, 30-Day and 1-year all-cause mortalities between sexes with adjustments for other covariates based on 5 models. The following steps were used to model in-hospital, 30-Day and 1-year mortalities (dependent, outcome variable) and sex (independent, predictor variable): in model 1, only sex was entered as the unadjusted predictor variable. In model 2, the following covariates were adjusted based on a step-wise approach: age, admission heart rate, admission systolic pressure, Killip class IV at presentation and elevated creatinine kinase. In model 3, additional covariates (coronary risk factors) were entered: cigarette smoking, DM, hypertension and ethnicity. In model 4, hospital management factors were added to the existing covariates: PCI, CABG and in-hospital use of aspirin, beta-blocker, angiotensin-converting enzyme inhibitor (ACE-I) and statin. Finally, in model 5, the participating centers (institutional factors) were added to the covariates.

The results were reported as odds ratio (OR) with 95% confidence intervals (CI) for sex differences. A p value of <0.05 was considered statistically significant. All calculations were performed using STATA software (version 16.1, Stata-Corp LLC, Texas, USA).

## Results

### Baseline characteristics and risk factors

We analysed the data of 35,232 ACS patients from 1st. Jan 2012 through 31st. Dec 2016 (Fig 1). Of these 27,989 (79.44%) were men and 7,243 (20.56%) were women with a men-to-women ratio of 4:1. Across all the ACS groups there were more men than women, more so in the STEMI cohort (S1 Fig).

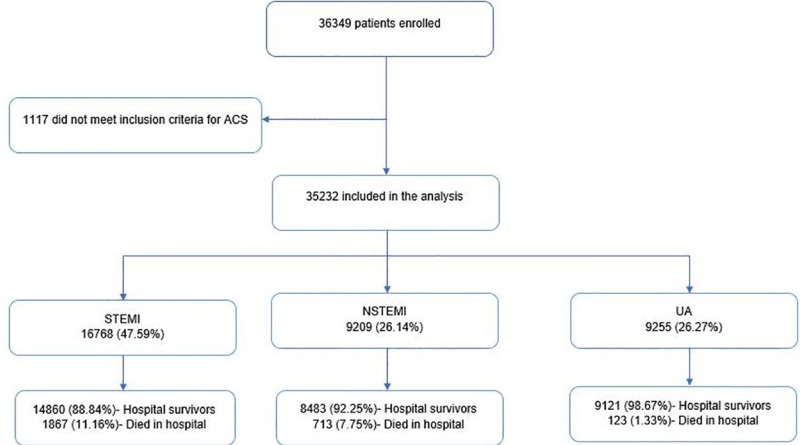

**Fig 1. NCVD-ACS registry flowchart, 2012–2016.** NSTEMI, non-ST-segment elevation myocardial infarction; STEMI, ST-segment elevation myocardial infarction; UA, unstable angina.

**Table 1. Baseline characteristics and risk factors for NCVD-ACS, 2012–2016 (n = 35232).**

| | STEMI | | | NSTEMI | | | UA | | |
|---|---|---|---|---|---|---|---|---|---|
| | **Men** | **Women** | *p* **value** | **Men** | **Women** | *p* **value** | **Men** | **Women** | *p* **value** |
| Age, yrs | 55.13 ± 11.61 | 63.41 ± 11.78 | < 0.001* | 59.29 ± 11.88 | 65.68 ± 11.37 | < 0.001* | 59.04 ± 11.66 | 63.47 ± 11.88 | < 0.001* |
| Ethnic group | | | | | | | | | |
| Malay | 8107 (56.16) | 1230 (52.74) | 0.002 | 3351 (47.26) | 917 (43.30) | 0.001 | 2992 (46.30) | 1128 (40.39) | < 0.001 |
| Non- Malay† | 6329 (43.84) | 1102 (47.26) | | 3740 (52.74) | 1201 (56.70) | | 3470 (53.70) | 1665 (59.61) | |
| Smoker | | | | | | | | | |
| current, former | 10823 (78.22) | 218 (9.77) | < 0.001 | 4615 (69.70) | 159 (7.93) | < 0.001 | 3940 (65.66) | 144 (5.52) | < 0.001 |
| Diabetes | 4786 (33.15) | 1296 (55.57) | < 0.001 | 3179 (44.83) | 1317 (62.18) | < 0.001 | 2862 (44.29) | 1582 (56.64) | < 0.001 |
| Hypertension | 6615 (45.82) | 1591 (68.22) | < 0.001 | 4581 (64.60) | 1698 (80.17) | < 0.001 | 4514 (69.85) | 2222 (79.56) | < 0.001 |
| Dyslipidemia | 3321 (23.00) | 660 (28.30) | < 0.001 | 2828 (39.88) | 917 (43.30) | 0.008 | 3214 (49.74) | 1469 (52.60) | 0.02 |
| Previous MI | 1525 (10.56) | 189 (8.10) | < 0.001 | 1395 (19.67) | 298 (14.07) | < 0.001 | 1526 (23.61) | 414 (14.82) | < 0.001 |
| Previous heart failure | 289 (2.00) | 95 (4.07) | < 0.001 | 606 (8.55) | 244 (11.52) | < 0.001 | 432 (6.69) | 202 (7.23) | 0.452 |
| Previous renal failure | 441 (3.05) | 127 (5.45) | < 0.001 | 797 (11.24) | 326 (15.39) | < 0.001 | 587 (9.08) | 279 (9.99) | 0.213 |
| CVA | 366 (2.54) | 105 (4.50) | < 0.001 | 309 (4.36) | 107 (5.05) | 0.04 | 287 (4.44) | 120 (4.30) | 0.59 |
| PVD | 32 (0.22) | 7 (0.30) | < 0.001 | 55 (0.78) | 17 (0.80) | 0.378 | 46 (0.71) | 20 (0.72) | 0.584 |
| Heart rate, beat/min | 82.57 ± 21.39 | 86.35 ± 22.58 | < 0.001* | 85.34 ± 21.76 | 90.02 ± 23.37 | < 0.001* | 79.63 ± 18.32 | 82.11 ± 18.65 | < 0.001* |
| SBP, mmHg | 132.63 ± 28.32 | 134.54 ± 31.72 | 0.003* | 139.50 ± 28.80 | 143.35 ± 31.70 | < 0.001* | 142.13 ± 25.79 | 147.95 ± 27.48 | < 0.001* |
| BMI, kg/m² | 26.12 ± 4.21 | 26.02 ± 4.77 | 0.571* | 26.06 ± 4.18 | 26.16 ± 5.23 | 0.562* | 26.05 ± 4.17 | 26.55 ± 5.08 | 0.001* |
| Killip class | | | | | | | | | |
| I (No heart failure) | 8587 (59.48) | 1233 (52.87) | < 0.001 | 3565 (50.27) | 952 (44.95) | < 0.001 | 2914 (45.09) | 1177 (42.14) | 0.069 |
| II (Heart failure) | 2479 (17.17) | 384 (16.47) | | 828 (11.68) | 300 (14.16) | | 353 (5.46) | 189 (6.77) | |
| III (Pulmonary edema) | 534 (3.70) | 155 (6.65) | | 380 (5.36) | 149 (7.03) | | 73 (1.16) | 34 (1.22) | |
| IV (Cardiogenic shock) | 2071 (14.35) | 427 (18.31) | | 349 (4.92) | 149 (7.03) | | 48 (0.74) | 21 (0.75) | |

Values are mean ± SD or %.

BMI, body mass index; CVA, cerebrovascular accident; MI, myocardial infarction; NCVD-ACS, National Cardiovascular Disease Database—Acute Coronary Syndrome; NSTEMI.

non-ST-segment elevation myocardial infarction; PVD, peripheral vascular disease; SBP, systolic blood pressure; STEMI, ST-segment elevation myocardial infarction; UA, unstable angina.

*Independent t test for differences between 2 means. All categorical variables are expressed as percentages. Chi-square test for 2 x 2 table (using Fisher exact test) for categorical variables.

†Non-Malay: Chinese, Indian, Indigenous (Orang Asli), and minor ethnic groups.

In all ACS groups of STEMI, NSTEMI and UA, women were significantly older (63.41 ± 11.78 vs. 55.13 ± 11.61, p<0.001; 65.68 ± 11.37 vs. 59.29 ± 11.88, p<0.001; 63.47 ± 11.88 vs. 59.04 ± 11.66, p<0.001, respectively). Women were also more likely to have DM, hypertension, dyslipidemia, previous heart failure and renal failure than men (Table 1). Cerebrovascular accident (CVA) was more common in women presenting with STEMI and NSTEMI than men.

Women were less likely to be former or current smokers or have a previous history of myocardial infarction than men in all ACS groups. At presentation, women in all ACS groups had higher heart rates and systolic blood pressure than men. Women also presented with significantly higher Killip classes in the STEMI and NSTEMI cohorts compared to men.

## In-hospital medications received

On admission most patients were prescribed antiplatelets and statin (Table 2). Overall, more than 90% of both sexes received either aspirin or another antiplatelet while 88.77% received

**Table 2. In-hospital medications, 2012–2016.**

| | STEMI | | | NSTEMI | | | UA | | |
|---|---|---|---|---|---|---|---|---|---|
| | **Men** | **Women** | ***p*** **value*** | **Men** | **Women** | ***p*** **value*** | **Men** | **Women** | ***p*** **value*** |
| Antiplatelets | | | | | | | | | |
| Aspirin | 13725 (95.07) | 2191 (93.95) | 0.048 | 6692 (94.37) | 1989 (93.91) | 0.618 | 6059 (93.76) | 2574 (92.16) | 0.015 |
| Other antiplatelets | 13791 (95.53) | 2222 (95.28) | 0.062 | 6683 (94.25) | 2002 (94.52) | 0.846 | 6008 (92.97) | 2554 (91.44) | 0.027 |
| Anticoagulants | | | | | | | | | |
| Heparin | 2248 (15.57) | 330 (14.15) | 0.078 | 542 (7.64) | 137 (6.47) | 0.051 | 272 (4.21) | 91 (3.26) | 0.015 |
| LMWH | 2573 (17.82) | 458 (19.64) | 0.073 | 1843 (25.99) | 653 (30.83) | < 0.001 | 1218 (18.85) | 546 (19.55) | 0.476 |
| Antihypertensives | | | | | | | | | |
| Beta-blocker | 7703 (53.36) | 1168 (50.09) | 0.002 | 4179 (58.93) | 1152 (54.39) | < 0.001 | 4143 (64.11) | 1752 (62.73) | 0.329 |
| ACE-I | 6325 (43.81) | 944 (40.48) | 0.003 | 3381 (47.68) | 857 (40.46) | < 0.001 | 3382 (52.34) | 1413 (50.59) | 0.23 |
| ARB | 305 (2.11) | 77 (3.30) | 0.001 | 400 (5.64) | 171 (8.07) | < 0.001 | 562 (8.70) | 318 (11.39) | < 0.001 |
| Diuretics | 2736 (18.95) | 577 (24.74) | < 0.001 | 2109 (29.74) | 833 (39.33) | < 0.001 | 1461 (22.61) | 715 (25.60) | 0.007 |
| CCB | 571 (3.96) | 159 (6.82) | < 0.001 | 951 (13.41) | 396 (18.70) | < 0.001 | 1210 (18.72) | 679 (24.31) | < 0.001 |
| Antidiabetic agents | | | | | | | | | |
| OHA | 2559 (17.73) | 677 (29.03) | < 0.001 | 1737 (24.50) | 682 (32.20) | < 0.001 | 1711 (26.48) | 893 (31.97) | < 0.001 |
| Insulin | 2911 (20.16)) | 861 (36.92) | < 0.001 | 1527 (21.53) | 723 (34.14) | < 0.001 | 1046 (16.19) | 620 (22.20) | < 0.001 |
| Statins | | | | | | | | | |
| Statin | 13026 (90.23) | 2046 (87.74) | < 0.001 | 6244 (88.06) | 1849 (87.30) | 0.566 | 5868 (90.81) | 2522 (90.30) | 0.727 |

Values are n (%).

ACE-I, angiotensin-converting enzyme inhibitor; ARB, angiotensin receptor blocker; CCB, calcium channel blocker; LMWH, low molecular weight heparin; OHA, oral hypoglycemic agents; other abbreviations as in Table 1.

*Chi-square test for 2 x 2 table (using Fisher exact test) for categorical variables. All categorical variables are expressed as number (%).

dual antiplatelets, although fewer women were prescribed aspirin in the STEMI and UA cohorts. Across the ACS stratum, more than 50% were prescribed angiotensin-converting enzyme-inhibitor (ACE-I) and more than 40% were prescribed beta-blocker for both sexes. However, women were less likely to receive beta-blocker and ACE-I in the STEMI and NSTEMI cohorts. More than 87% of both sexes received statin but women in the STEMI cohort were less likely to receive one compared to men. On the contrary, use of angiotensin receptor blocker (ARB), diuretics, calcium channel blocker (CCB), oral hypoglycaemic agents (OHA) and insulin were significantly higher in women across all ACS groups.

## Invasive therapeutic procedures, culprit arteries and numbers of diseased vessels

Across the ACS stratum of STEMI, NSTEMI and UA, women were less likely to undergo coronary angiography (40.91% vs. 48.93%, p< 0.001; 28% vs. 37%, p< 0.001; 17.83 vs. 21.96, p<0.001, respectively) and PCI (32.03% vs. 40.04%, p< 0.001; 16.43% vs. 21.55%, p<0.001; 8.23% vs. 10.79%, p = 0.001, respectively) than men (Table 3). With regards to culprit artery, across the ACS stratum, the left anterior descending artery (LAD) was the most commonly affected vessel (accounting for > 46%) followed by right coronary artery, left circumflex (LCx) and left main stem (LMS) for both sexes. There was a significant association between culprit artery and sex of patient among STEMI patients (p = 0.005) and number of diseased vessel and sex of patients among NSTEMI patients (p = 0.007). Left main stem disease was less common in women in the STEMI cohort, but was more common in women in the NSTEMI and UA

**Table 3. In hospital procedures, culprit artery, number of disease vessels, and outcomes, 2012–2016.**

| | STEMI | | | NSTEMI | | | UA | | |
|---|---|---|---|---|---|---|---|---|---|
| | **Men** | **Women** | ***p* value** | **Men** | **Women** | ***p* value** | **Men** | **Women** | ***p* value** |
| Angiography | 7063 (48.93) | 954 (40.91) | < 0.001 | 2624 (37.00) | 603 (28.47) | < 0.001 | 1419 (21.96) | 498 (17.83) | < 0.001 |
| PCI | 5780 (40.04) | 747 (32.03) | < 0.001 | 1528 (21.55) | 348 (16.43) | < 0.001 | 697 (10.79) | 230 (8.23) | 0.001 |
| CABG | 164 (1.14) | 15 (0.64) | 0.124 | 144 (2.03) | 29 (1.37) | 0.033 | 108 (1.67) | 26 (0.93) | 0.008 |
| Culprit artery | | | | | | | | | |
| LAD | 2646 (52.87) | 293 (46.36) | 0.005 | 808 (50.41) | 190 (52.92) | 0.707 | 480 (50.26) | 162 (51.10) | 0.393 |
| RCA | 1865 (35.26) | 285 (45.09) | | 423 (26.39) | 95 (26.46) | | 272 (28.48) | 98 (30.91) | |
| LCX | 379 (7.57) | 43 (6.80) | | 279 (17.40) | 53 (14.76) | | 143 (15.97) | 40 (12.62) | |
| LMS | 103 (2.06) | 10 (1.58) | | 63 (3.93) | 16 (4.46) | | 37 (3.87) | 14 (4.42) | |
| Bypass graft | 12 (0.24) | 1 (0.16) | | 30 (1.87) | 5 (1.39) | | 23 (2.41) | 3 (0.95) | |
| No. of diseased vessels | | | | | | | | | |
| 0 | 18 (0.36) | 1 (0.16) | 0.686 | 11 (0.70) | 6 (1.65) | 0.007 | 10 (1.08) | 8 (2.52) | 0.142 |
| 1 | 3437 (68.17) | 420 (66.56) | | 963 (61.18) | 203 (55.92) | | 648 (69.90) | 206 (64.98) | |
| 2 | 985 (19.54) | 129 (20.44) | | 350 (22.24) | 73 (20.11) | | 157 (16.94) | 63 (19.87) | |
| 3 | 602 (11.94) | 81 (12.84) | | 250 (15.88) | 81 (22.31) | | 112 (12.08) | 40 (12.62) | |
| Outcomes | | | | | | | | | |
| CCU days | 3.07 ± 2.59 | 3.21 ± 2.78 | 0.057* | 3.69 ± 3.38 | 4.23 ± 4.10 | 0.030* | 3.28 ± 2.88 | 3.28 ± 3.12 | 0.985* |
| Total days | 5.72 ± 5.57 | 6.25 ± 6.61 | 0.001* | 6.38 ± 7.57 | 6.84 ± 7.15 | 0.015* | 4.75 ± 5.81 | 4.89 ± 5.86 | 0.298* |
| Bleeding (TIMI†) | | | | | | | | | |
| Major | 26 (0.66) | 7 (1.03) | 0.004 | 10 (0.49) | 1 (0.17) | 0.664 | 4 (0.23) | 3 (0.39) | 0.887 |
| Minor | 149 (3.75) | 18 (2.64) | | 51 (2.50) | 15 (2.62) | | 34 (1.92) | 16 (2.09) | |
| Minimal | 69 (1.74) | 27 (3.96) | | 31 (1.52) | 12 (2.10) | | 35 (1.98) | 19 (2.48) | |
| None | 3433 (86.50) | 573 (84.14) | | 1611 (78.85) | 436 (76.22) | | 1552 (87.68) | 663 (86.67) | |
| In-hospital mortality ‡ | 1411 (9.80) | 456 (19.63) | < 0.001 | 503 (7.10) | 210 (9.93) | < 0.001 | 83 (1.29) | 40 (1.43) | 0.568 |
| 30-day mortality ‡ | 1640 (13.53) | 533 (26.80) | < 0.001 | 670 (11.24) | 287 (16.02) | < 0.001 | 170 (3.02) | 71 (2.93) | 0.83 |
| 1-year mortality ‡ | 2166 (19.07) | 698 (37.39) | < 0.001 | 1300 (23.22) | 520 (30.70) | < 0.001 | 590 (11.04) | 277 (12.19) | 0.149 |

Values are n (%) or mean ± SD.

CABG, coronary artery bypass graft; CCU, coronary care unit; LAD, left anterior descending artery; LCX, left circumflex artery; LMA, left main artery; PCI, percutaneous coronary intervention; RCA, right coronary artery; TIMI, Thrombolysis in Myocardial Infarction; other abbreviations as in Table 1.

*Independent t test for differences between 2 means. Chi-square test for 2 x 2 table (using Fisher exact test) for categorical variables. All categorical variables are expressed as n (%).

† TIMI major bleeding involves a haemoglobin drop >5 g/dl (with or without an identified site) or intracranial haemorrhage or cardiac tamponade.

‡ Unadjusted in-hospital, 30-day and 1-year all-cause mortality rate.

cohorts compared to men. Across the ACS stratum of STEMI, NSTEMI and UA, over 55% had single vessel disease for both sexes.

## Treatment of STEMI

More than 76% of both sexes received some form of reperfusion either by fibrinolysis or primary PCI, although more received fibrinolysis than PCI (S1 Table). More women did not receive any form of reperfusion therapy (23.01% vs. 15.71%, p<0.001, respectively) and a lower proportion received primary PCI (12.97% vs. 14.48%, respectively) or fibrinolysis (64.03% vs. 69.81%, respectively) than men. Women had significantly longer median door-to-needle time (59.0 min [IQR: 90] vs. 45.0 min [IQR: 65], respectively, p = 0.001) and door-to-balloon time (95.0 min [IQR: 105] vs. 78.0 min [IQR: 80.5], respectively, p = 0.003) compared to men.

## In-hospital clinical outcomes and mortality outcomes

Overall, the rates of major bleeding were low (0.17%-1.03%) among all ACS groups and both sexes (Table 3). There is a significant association between bleeding and sex of patient in the STEMI cohort (p = 0.004) with women having more major bleeding (1.03% vs. 0.66%, respectively) than men. Women had longer coronary care unit and total hospital stay than men in the STEMI (3.21 ± 2.78 days vs. 3.07 ± 2.59 days, p = 0.057; 6.25 ± 6.61 days vs. 5.72 ± 5.57 days, p = 0.001, respectively) and NSTEMI (4.23 ± 4.10 days vs. 3.69 ± 3.38 days, p = 0.030; 6.84 ± 7.15 vs. 6.38 ± 7.57 days, p = 0.015, respectively) cohorts (Table 3).

**In-hospital mortality.** In-hospital mortality increases with age, women had consistently higher mortality in all age groups below 70 years, while above 70 years mortality were higher in men [S2 Fig]. Overall, women had higher in-hospital mortality rates than men.

The unadjusted in-hospital mortality rates of both sexes were higher in the STEMI cohort than the NSTEMI and UA cohorts (Table 3). Women had higher unadjusted in-hospital mortality rates compared to men for the STEMI (19.63% vs. 9.80%, respectively, p<0.001; unadjusted OR: 2.25, 95% CI: 2.00–2.53, p<0.001) and NSTEMI (9.93% vs. 7.10%, respectively, p<0.001; unadjusted OR: 1.44, 95% CI: 1.22–1.71, p<0.001) cohorts (Tables 3 and 4). The unadjusted in-hospital mortality rates for UA were however not significantly different between the sexes (1.43% vs. 1.29%, respectively, p = 0.568; unadjusted OR: 1.12, 95% CI: 0.76–1.63, p = 0.568).

In the STEMI cohort, after multiple logistic regression analyses including age, clinical features at presentation (admission heart rate, admission systolic blood pressure, Killip class IV at presentation, elevated creatinine kinase) and coronary risk factors (cigarette smoking, DM, hypertension and ethnicity), the in-hospital mortality difference between women and men were no longer significant (adjusted OR: 1.19, 95% CI: 0.97–1.46, p = 0.09) (Table 4). In fact, with further adjustments of hospital management and institutional factors the difference was further reduced (adjusted OR: 1.10, 95% CI: 0.89–1.37, p = 0.376). Meanwhile, for the NSTEMI cohort, after adjusting for age and clinical presentation (admission heart rate, admission systolic blood pressure, Killip class IV at presentation, elevated creatinine kinase), the in-hospital mortality difference between women and men were no longer significant (adjusted OR: 1.04, 95% CI: 0.81–1.34, p = 0.758). This difference was further reduced when adjustments were made to include coronary risk factors, hospital management and institutional factors (adjusted OR: 0.74, 95% CI: 0.53–1.02, p = 0.065).

For women prognostic factors that increase in-hospital mortality include presentation with Killip class IV (HR: 5.87, 95% CI: 3.76–9.16, p<0.001), elevated creatinine kinase (HR: 1.93, 95% CI: 1.19–3.11, p = 0.007) and age (HR: 1.02, 95% CI:1.00–1.04, p = 0.034), whereas prognostic factors that decrease in-hospital mortality were use of ACE-I (HR: 0.23, 95% CI:0.10–0.52, p<0.001), statin (HR: 0.44, 95% CI 0.26–0.72, p = 0.001) and beta-blocker (HR: 0.56, 95% CI:0.34–0.94, p = 0.029) (S2 Table). Meanwhile, for men the prognostic factors that increase in-hospital mortality were Killip class IV at presentation (HR: 2.71, 95% CI:2.05–3.58, p< 0.001), elevated creatinine kinase (HR: 1.71, 95% CI:0.53–2.35, p = 0.001), age (HR: 1.03, 95% CI:1.02–1.04, p<0.001) and heart rate at presentation (HR: 1.02, 95% CI:1.01–1.02, p<0.001), whereas the prognostic factors that decrease in-hospital mortality were use of beta-blocker (HR: 0.29, 95% CI: 0.21–0.41, p<0.001), ACE-I (HR: 0.33, 95% CI: 0.22–0.50, p<0.001), aspirin (HR: 0.49, 95% CI: 0.28–0.86, p = 0.013) and statin (HR: 0.51, 95% CI: 0.36–0.71, p<0.001) and PCI (HR: 0.7, 95% CI: 0.53–0.92, p = 0.010).

**30-Day mortality.** The unadjusted 30-Day mortality for women remained higher compared to men in the STEMI (unadjusted OR: 2.34, 95% CI: 2.09–2.62, p<0.001) and NSTEMI (unadjusted OR: 1.51, 95% CI: 1.30–1.75, p<0.001) cohorts. However, the unadjusted 30-Day

**Table 4. Result of multiple logistic regression analyses of in-hospital mortality, 30-day mortality and 1-year mortality, 2012–2016.**

| Outcomes | Models | STEMI | p value | NSTEMI | p value | UA | p value |
|---|---|---|---|---|---|---|---|
| In-hospital mortality | Model 1 | 2.25 (2.00, 2.53) | < 0.001 | 1.44 (1.22–1.71) | < 0.001 | 1.12 (0.76–1.63) | 0.568 |
| | Model 2 | 1.43 (1.21–1.68) | < 0.001 | 1.04 (0.81–1.34) | 0.758 | 1.02 (0.59–1.75) | 0.952 |
| | Model 3 | 1.19 (0.97–1.46) | 0.09 | 0.91 (0.66–1.23) | 0.528 | 1.11 (0.57–2.15) | 0.768 |
| | Model 4 | 1.11 (0.90–1.38) | 0.333 | 0.80 (0.58–1.10) | 0.164 | 1.01 (0.51–2.00) | 0.974 |
| | Model 5 | 1.10 (0.89–1.37) | 0.376 | 0.74 (0.53–1.02) | 0.065 | 0.92 (0.46–1.83) | 0.815 |
| 30-day mortality | Model 1 | 2.34 (2.09–2.62) | < 0.001 | 1.51 (1.30–1.75) | < 0.001 | 0.97 (0.73–1.29) | 0.83 |
| | Model 2 | 1.50 (1.28–1.75) | < 0.001 | 1.04 (0.83–1.31) | 0.714 | 0.87 (0.57–1.34) | 0.528 |
| | Model 3 | 1.28 (1.05–1.55) | 0.013 | 0.90 (0.68–1.18) | 0.431 | 1.17 (0.69–2.00) | 0.561 |
| | Model 4 | 1.24 (1.00–1.52) | 0.047 | 0.83 (0.63–1.10) | 0.185 | 1.11 (0.65–1.92) | 0.696 |
| | Model 5 | 1.22 (0.99–1.51) | 0.059 | 0.78 (0.59–1.03) | 0.079 | 1.05 (0.61–1.81) | 0.869 |
| 1-year mortality | Model 1 | 2.53 (2.28–2.81) | < 0.001 | 1.46 (1.30–1.65) | < 0.001 | 1.12 (0.96–1.30) | 0.149 |
| | Model 2 | 1.63 (1.42–1.88) | < 0.001 | 1.01 (0.84–1.21) | 0.929 | 0.85 (0.66–1.10) | 0.209 |
| | Model 3 | 1.36 (1.14–1.61) | < 0.001 | 0.90 (0.73–1.11) | 0.33 | 0.90 (0.66–1.23) | 0.498 |
| | Model 4 | 1.32 (1.10–1.58) | 0.002 | 0.87 (0.70–1.09) | 0.223 | 0.89 (0.65–1.22) | 0.472 |
| | Model 5 | 1.31 (1.09–1.57) | 0.003 | 0.84 (0.67–1.04) | 0.116 | 0.84 (0.61–1.16) | 0.287 |

Values are OR (95% CI). Estimate for sex (women) was adjusted for other covariates.

Model 1: Unadjusted logistic regression analysis including sex only (men as reference).

Model 2: Logistic regression analysis adjusted for age, admission HR, admission SBP, Killip class IV at presentation, elevated creatinine kinase (adapted from Steg et al. [19]).

Model 3: Model 2 plus cigarette smoking, DM, hypertension, ethnicity (coronary risk factors).

Model 4: Model 3 plus PCI, CABG, in-hospital use of aspirin, beta-blocker, ACE-I, statin (hospital management).

Model 5: Model 4 plus participating centers (institutions).

CI, confidence interval; DM, diabetes mellitus; HR, heart rate; OR, odds ratio; other abbreviations as in Tables 1–3.

mortality rates for UA were not significantly different between the sexes (unadjusted OR: 0.97, 95% CI: 0.73–1.29, p = 0.830) (Table 4).

In the STEMI cohort, the 30-Day mortality difference between women and men only became insignificant (adjusted OR: 1.22, 95% CI: 0.99–1.51, p = 0.059) after multiple logistic regression analyses including age, clinical features at presentation (admission heart rate, admission systolic blood pressure, Killip class IV at presentation, elevated creatinine kinase), coronary risk factors (cigarette smoking, DM, hypertension and ethnicity), hospital management (percutaneous coronary intervention, CABG, in-hospital use of aspirin, beta-blocker, ACE-I and statin) and participating center (institutional factors). Meanwhile, for the NSTEMI cohort, after adjusting for age and clinical presentation (admission heart rate, admission systolic blood pressure, Killip class IV at presentation, elevated creatinine kinase), the 30-Day mortality difference between women and men were no longer significant (adjusted OR: 1.04, 95% CI: 0.83–1.31, p = 0.714). This difference was further reduced when adjustments were made to include coronary risk factors, hospital management and institutional factors (adjusted OR: 0.78, 95% CI: 0.59–1.03, p = 0.079).

**1-year mortality.** The unadjusted 1-year mortality for women remained higher compared to men in the STEMI (unadjusted OR: 2.53, 95% CI: 2.28–2.81, p<0.001) and NSTEMI (unadjusted OR: 1.46, 95% CI: 1.30–1.65, p<0.001) cohort. However, the unadjusted 1-year mortality rates for UA were not significantly different between the sexes (unadjusted OR: 1.12, 95% CI: 0.96–1.30, p = 0.149) (Table 4).

In the STEMI cohort, at 1 year despite adjustments for all covariates including age, clinical presentation, coronary risk factors, hospital management and institutional factors, the 1-year mortality for women remained higher than men (adjusted OR: 1.31, 95% CI: 1.09–1.57, p = 0.003). Meanwhile, for the NSTEMI cohort, after adjusting for age and clinical presentation the 1-year mortality difference between women and men were no longer significant (adjusted OR: 1.01, 95% CI: 0.84–1.21, p = 0.929). This difference was further reduced when adjustments were made to include coronary risk factors, hospital management and institutional factors (adjusted OR: 0.84, 95% CI: 0.67–1.04, p = 0.116).

## Observed trends between 2006–2010 and 2012–2016

**Baseline characteristics and risk factors.** The current cohort (2012–2016: period 2) seen an increase in percentage of men presenting with ACS, 79.44% were men and 20.56% were women, with a men-to-women ratio of 4:1. In contrast to the data from March 2006 to February 2010 (period 1), which included 13,591 ACS patients, of which 10,299 (75.8%) were men and 3,292 (24.2%) were women with a men-to-women ratio of 3:1. This male dominance is seen across the ACS stratum for both periods [S3 Fig].

For both periods, in all ACS groups, women were generally older and more likely to have DM, hypertension, previous heart failure and renal failure than men [Fig 2]. However, unlike period 1, dyslipidemia has now become more common in women. Cerebrovascular accident which was more common in women presenting with STEMI and NSTEMI during period 1 is now more common only in the STEMI cohort.

Women were less likely to be former or current smokers or have a previous history of myocardial infarction in all ACS groups in period 2. This is unlike period 1 where women were less likely to have previous history of myocardial infarction only in NSTEMI and UA cohorts. Of note, the percentages of patients with diabetes mellitus, hypertension, dyslipidemia, previous myocardial infarction and previous heart failure in all ACS groups have decrease for both sexes whereas only the percentage of women smokers has decrease during period 2 [Fig 2].

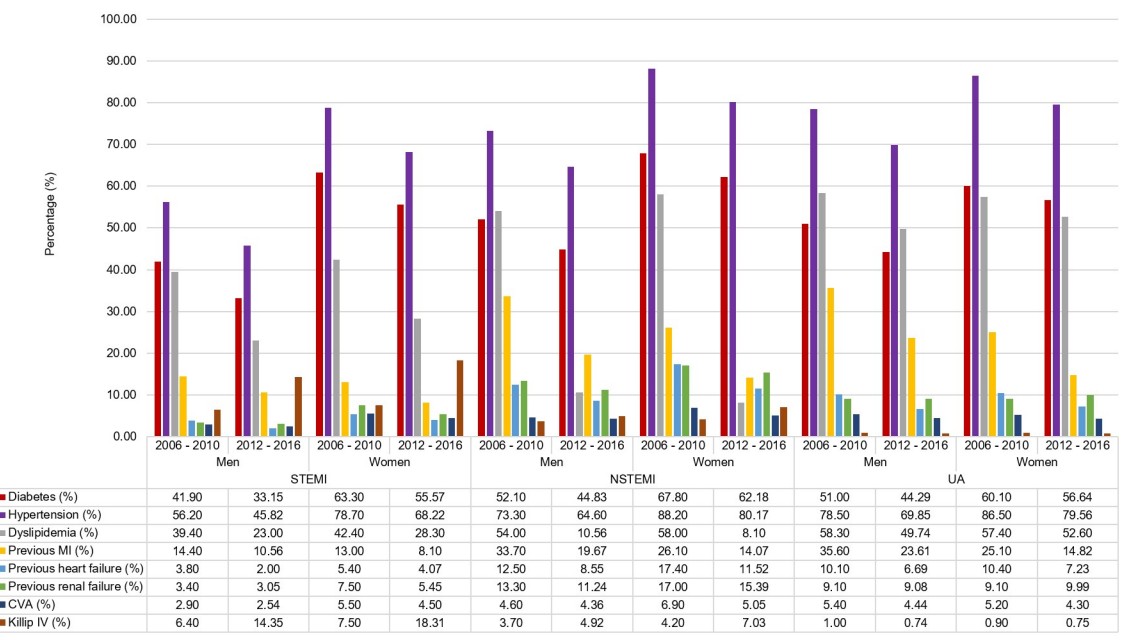

**Fig 2. Baseline characteristics & risk factors comparing 2006–2010 and 2012–2016.**

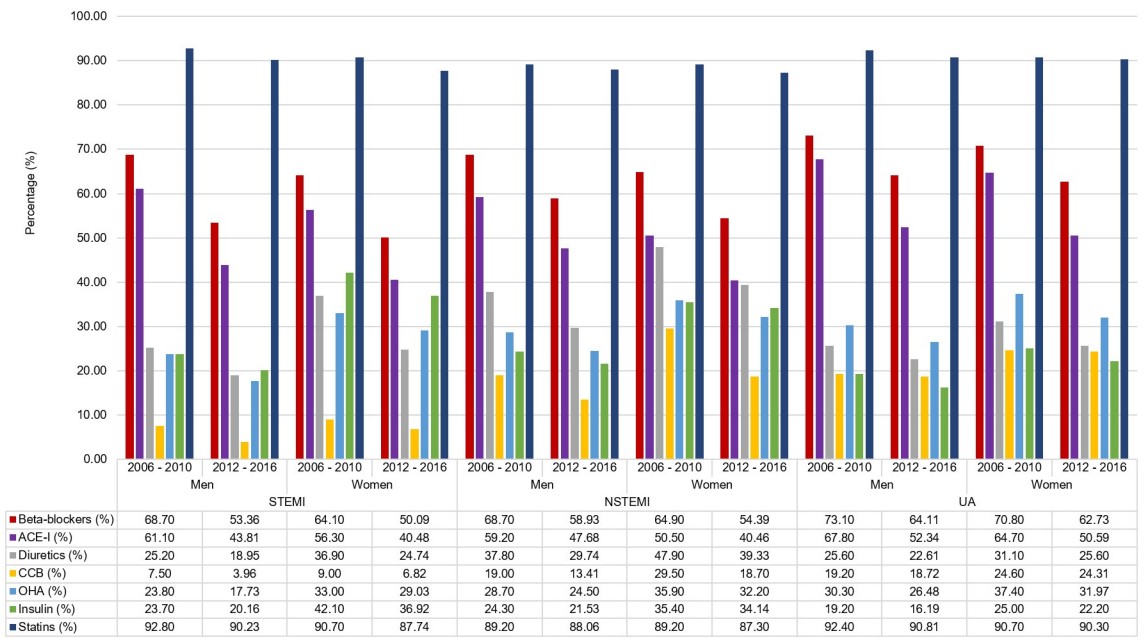

| | STEMI | | | | NSTEMI | | | | UA | | | |
| | Men | | Women | | Men | | Women | | Men | | Women | |
| | 2006 – 2010 | 2012 – 2016 | 2006 – 2010 | 2012 – 2016 | 2006 – 2010 | 2012 – 2016 | 2006 – 2010 | 2012 – 2016 | 2006 – 2010 | 2012 – 2016 | 2006 – 2010 | 2012 – 2016 |
|---|---|---|---|---|---|---|---|---|---|---|---|---|
| Beta-blockers (%) | 68.70 | 53.36 | 64.10 | 50.09 | 68.70 | 58.93 | 64.90 | 54.39 | 73.10 | 64.11 | 70.80 | 62.73 |
| ACE-I (%) | 61.10 | 43.81 | 56.30 | 40.48 | 59.20 | 47.68 | 50.50 | 40.46 | 67.80 | 52.34 | 64.70 | 50.59 |
| Diuretics (%) | 25.20 | 18.95 | 36.90 | 24.74 | 37.80 | 29.74 | 47.90 | 39.33 | 25.60 | 22.61 | 31.10 | 25.60 |
| CCB (%) | 7.50 | 3.96 | 9.00 | 6.82 | 19.00 | 13.41 | 29.50 | 18.70 | 19.20 | 18.72 | 24.60 | 24.31 |
| OHA (%) | 23.80 | 17.73 | 33.00 | 29.03 | 28.70 | 24.50 | 35.90 | 32.20 | 30.30 | 26.48 | 37.40 | 31.97 |
| Insulin (%) | 23.70 | 20.16 | 42.10 | 36.92 | 24.30 | 21.53 | 35.40 | 34.14 | 19.20 | 16.19 | 25.00 | 22.20 |
| Statins (%) | 92.80 | 90.23 | 90.70 | 87.74 | 89.20 | 88.06 | 89.20 | 87.30 | 92.40 | 90.81 | 90.70 | 90.30 |

**Fig 3. In-hospital medications comparing 2006–2010 & 2012–2016.**

At presentation, women in all ACS groups had higher heart rates and systolic blood pressure than men for both periods. However, women in all ACS groups had higher Killip classes in period 2 compared to only in the STEMI and NSTEMI cohorts during period 1 [Fig 2]. Of note is the marked increase in the percentage of women compared to men presenting with Killip class IV in period 2 compared to period 1 in STEMI (18.31% vs. 14.35%, p<0.001; 7.5% vs. 6.4%, p-0.000, respectively) and NSTEMI (7.03% vs. 4.92%, p<0.001; 4.2 vs. 3.7%, p = 0.002, respectively).

## In-hospital medications received

The current cohort (period 2) was less likely to receive beta-blocker, ACE-I, diuretics, CCB, oral hypoglycaemic agents, insulin, and statin compared to period 1 for both sexes [Fig 3]. Of particular importance were the 3 medicines that has prognostic significance in ACS for women and men viz. beta-blocker, ACE-I and statin. For the STEMI cohort, the prescription rates for women and men between period 2 and 1 for beta-blocker were (50.09% vs. 53.36%, p = 0.002; 64.1% vs. 68.7%, p = 0.006, respectively), for ACE-I were (40.48% vs. 43.81%, p = 0.002; 56.3% vs. 61.1%, p = 0.008, respectively) and for statin were (87.74% vs. 90.23%, p< 0.001; 90.7% vs. 92.8%, p = 0.029 respectively) [Fig 3]. Likewise, in the NSTEMI cohort, the prescription rates for women and men between period 2 and 1 for beta-blocker were (54.39% vs. 58.93%, p<0.001; 64.9% vs. 68.7%, p = 0.023, respectively), for ACE-I were (40.46% vs. 47.68%, p = 0.001; 50.5% vs. 59.2%, p = 0.000, respectively) and for statin were (87.30% vs. 88.06% p< 0.566; 89.2% vs. 89.2%, p = 0.989, respectively).

## In-hospital procedures, culprit arteries and numbers of diseased vessels

There were marked increase in percentages of patients who underwent coronary angiography and PCI for both sexes across all ACS groups between period 2 compared to period 1, although the percentages for women were still less than men [Fig 4]. For angiography the percentage of

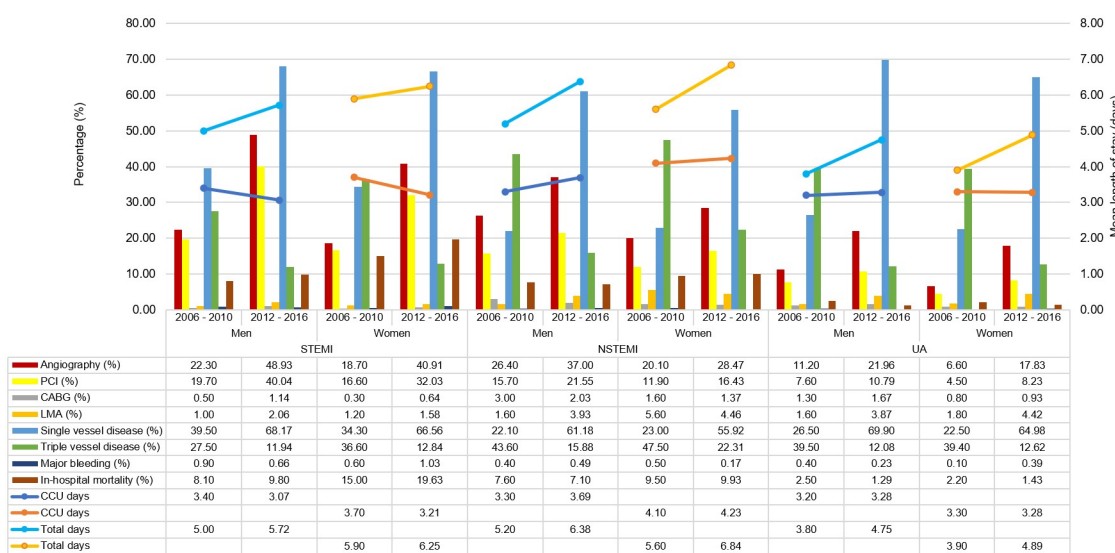

|  | STEMI | | | | NSTEMI | | | | UA | | | |
|  | Men | | Women | | Men | | Women | | Men | | Women | |
|  | 2006 - 2010 | 2012 - 2016 | 2006 - 2010 | 2012 - 2016 | 2006 - 2010 | 2012 - 2016 | 2006 - 2010 | 2012 - 2016 | 2006 - 2010 | 2012 - 2016 | 2006 - 2010 | 2012 - 2016 |
|---|---|---|---|---|---|---|---|---|---|---|---|---|
| Angiography (%) | 22.30 | 48.93 | 18.70 | 40.91 | 26.40 | 37.00 | 20.10 | 28.47 | 11.20 | 21.96 | 6.60 | 17.83 |
| PCI (%) | 19.70 | 40.04 | 16.60 | 32.03 | 15.70 | 21.55 | 11.90 | 16.43 | 7.60 | 10.79 | 4.50 | 8.23 |
| CABG (%) | 0.50 | 1.14 | 0.30 | 0.64 | 3.00 | 2.03 | 1.60 | 1.37 | 1.30 | 1.67 | 0.80 | 0.93 |
| LMA (%) | 1.00 | 2.06 | 1.20 | 1.58 | 1.60 | 3.93 | 5.60 | 4.46 | 1.60 | 3.87 | 1.80 | 4.42 |
| Single vessel disease (%) | 39.50 | 68.17 | 34.30 | 66.56 | 22.10 | 61.18 | 23.00 | 55.92 | 26.50 | 69.90 | 22.50 | 64.98 |
| Triple vessel disease (%) | 27.50 | 11.94 | 36.60 | 12.84 | 43.60 | 15.88 | 47.50 | 22.31 | 39.50 | 12.08 | 39.40 | 12.62 |
| Major bleeding (%) | 0.90 | 0.66 | 0.60 | 1.03 | 0.40 | 0.49 | 0.50 | 0.17 | 0.40 | 0.23 | 0.10 | 0.39 |
| In-hospital mortality (%) | 8.10 | 9.80 | 15.00 | 19.63 | 7.60 | 7.10 | 9.50 | 9.93 | 2.50 | 1.29 | 2.20 | 1.43 |
| CCU days | 3.40 | 3.07 |  |  | 3.30 | 3.69 |  |  | 3.20 | 3.28 |  |  |
| CCU days |  |  | 3.70 | 3.21 |  |  | 4.10 | 4.23 |  |  | 3.30 | 3.28 |
| Total days | 5.00 | 5.72 |  |  | 5.20 | 6.38 |  |  | 3.80 | 4.75 |  |  |
| Total days |  |  | 5.90 | 6.25 |  |  | 5.60 | 6.84 |  |  | 3.90 | 4.89 |

**Fig 4. In hospital procedures, culprit artery, number of diseased vessels & outcomes comparing 2006–2010 and 2012–2016.**

women compared to men between period 2 and period 1 for STEMI were (40.91% vs. 48.93%, p<0.001; 18.7% vs. 22.3%, p = 0.018, respectively), for NSTEMI were (28.47% vs. 37.00%, p <0.001; 20.1 vs. 26.4%, p = 0.000, respectively) and for UA were (17.83% vs. 21.96%, p<0.001; 6.6% vs. 11.2%, p = 0.000, respectively) (Fig 6). Whereas, for PCI the percentage of women compared to men between period 2 and period 1 for STEMI were (32.03% vs. 40.04%, p<0.001; 16.6% vs. 19.7%, p = 0.027, respectively), for NSTEMI were (16.437% vs. 21.55%, p <0.001; 11.9 vs. 15.7%, p = 0.003, respectively) and for UA were (8.23% vs. 10.79%, p = 0.001; 4.5% vs. 7.6%, p = 0.000, respectively).

The frequency of involvement of the LAD followed by RCA then LCx as culprit artery remained the same for both periods. However, there was a change in trend in culprit artery involving left main artery (LMA). Although the numbers are small, previously in period 1, it was more common in women across the ACS stratum; however, during period 2 it was more common in women only in the NSTEMI and UA group compared to men [Fig 4].

There was also a reversal in the percentages of number of diseased vessels. During period 1 there were more patients with triple vessel disease (average 41.5% for women, 36.9% for men) but for the current cohort there were more patients with single vessel disease (average 62.48% for women, 66.41% for men) [Fig 4].

## Treatment of STEMI

There was an increase in the percentage of patients receiving some form of reperfusion either by fibrinolysis or primary PCI between the current cohort (>76%) compared to period 1 (≥70%), although women still receive less reperfusion by either of these modality [Fig 5]. The increase was mainly due to the percentage of primary PCI which has doubled for both women and men between period 2 and period 1 (12.97 vs. 14.48, p<0.001; 6.2 vs. 6.7, p = 0.000, respectively). There were also improvement for both women and men in door-to-needle time (59.0 min [IQR: 90] vs. 45.0 min [IQR: 65], p = 0.001; 60.8 min [IQR: 87.9] vs. 49.7 min [IQR: 74.2], p = 0.000, respectively) and door-to balloon time (95.0 min [IQR: 105] vs. 78.0 min [IQR: 80.5], p = 0.003; 121 min [IQR: 109.0] vs. 110 min [IQR: 104.5], p = 0.244), respectively) between period 2 and period 1 [Fig 5].

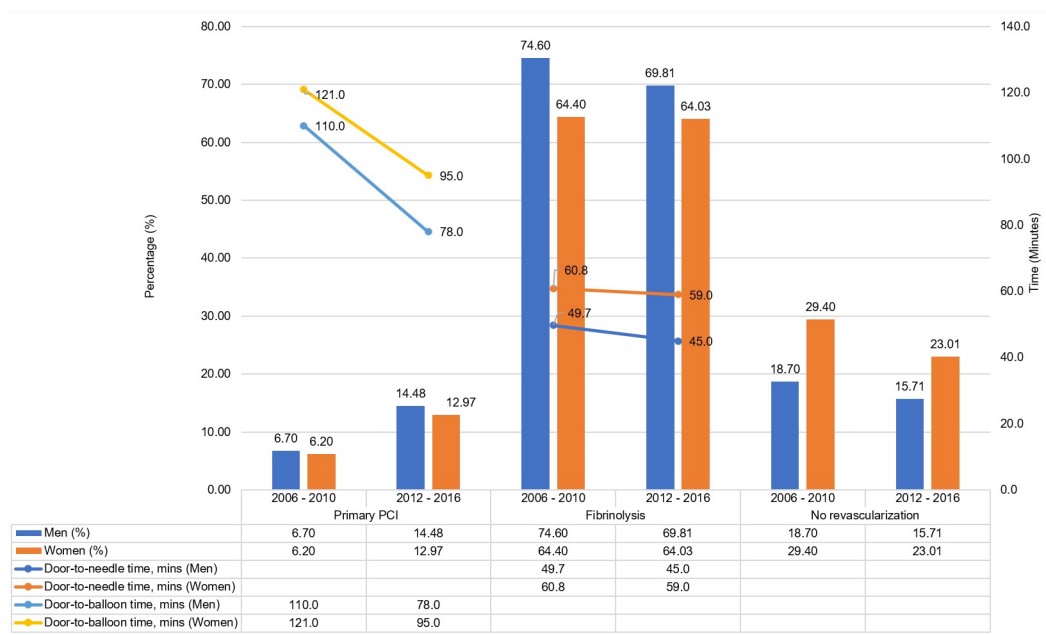

**Fig 5. Treatment of STEMI comparing 2006–2010 and 2012–2016.**

## In-hospital clinical outcomes and mortality

Although the rates of major bleeding were low, the percentage of women with major bleeding has increased whereas, it has decreased for men with STEMI in the current cohort compared to period 1 (1.03% vs. 0.66%, p = 0.004; 0.6% vs. 0.9%, p = 0.509, respectively) [Fig 4]. Where previously it was not significant, there is now a significant association between women and risk of bleeding.

Women had longer CCU and total hospital stay for both periods. However, the length of CCU stay for both sexes has decreased for STEMI cohort although the total length of hospital stay has increased. Whereas, both these parameters had increased for the NSTEMI cohort between periods 2 and 1 [Fig 4].

**In-hospital mortality.** Below the age of 70 years, although women consistently showed higher in-hospital mortality with advancing age in both time periods, the marked increase in mortality in women <40 years of age seen in the 2006–2010 cohort, were no longer evident now [Fig 6].

In the 2012–2016 cohort, the unadjusted in-hospital mortality rates were significantly higher in women for both the STEMI and NSTEMI groups. In contrast, for the 2006–2010 cohort it was only significantly higher in the STEMI group [Fig 4]. Of note, the in-hospital mortality rates had increased for both women and men in the STEMI cohort in period 2 compared to period 1 (19.63% vs. 9.80%, p<0.001; 15.0% vs. 8.1%, p = 0.000, respectively), despite the increase in rates of primary PCI and fibrinolysis. However, after adjusting for age, clinical features at presentation and coronary risk factors in the STEMI cohort for both time periods, the difference in in-hospital mortality between women and men were no longer significant. Likewise, for the NSTEMI cohort, after adjusting for age and clinical presentation the difference in in-hospital mortality between women and men were no longer significant.

**30-Day and 1-year mortality.** The previous analysis did not look at 30-day and 1-year mortality. As the current cohort (2012–2016) included a higher proportion of patients with

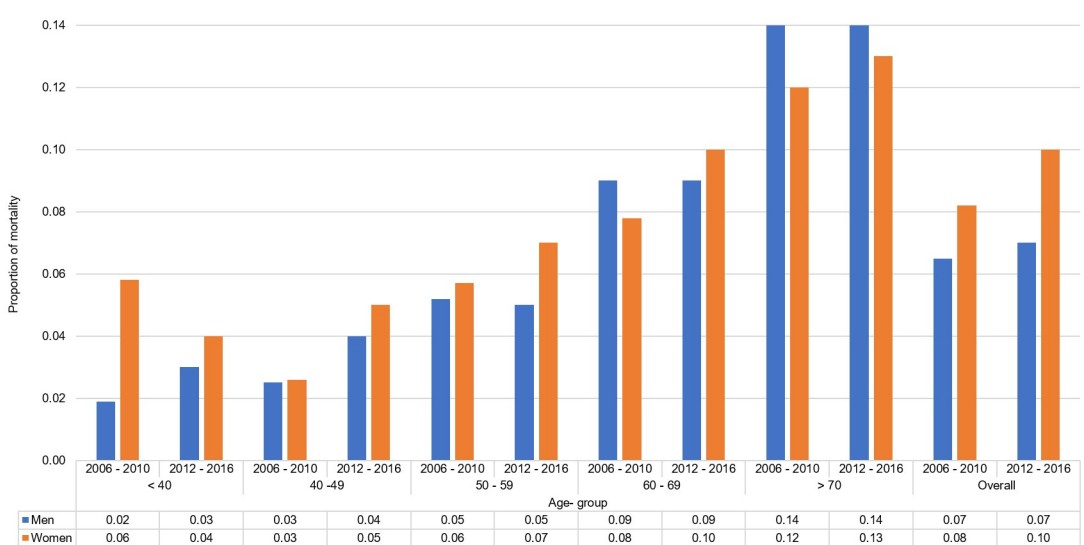

**Fig 6. Proportion of ACS in-hospital mortality by age groups and sex comparing 2006–2010 and 2012–2016.**

Killip class IV with consequent higher in-hospital mortality, we were interested to know how this cohort of patients would perform over time. The unadjusted 30-Day mortality for women remained higher compared to men in the STEMI and NSTEMI cohorts. In the STEMI cohort, the 30-Day mortality difference between women and men only became insignificant after multiple logistic regression analyses including age, clinical features at presentation, coronary risk factors, hospital management and participating center. Meanwhile, for the NSTEMI cohort, after adjusting for age and clinical presentation, the 30-Day mortality difference between women and men were no longer significant.

The unadjusted 1-year mortality for women remained higher compared to men in the STEMI and NSTEMI cohorts. In the STEMI cohort, at 1 year despite adjustments for all covariates including age, clinical presentation, coronary risk factors, hospital management and institutional factors, the 1-year mortality for women remained higher than men (adjusted OR:1.31, 95% CI: 1.09–1.57, p = 0.003). Meanwhile, for the NSTEMI cohort, after adjusting for age and clinical presentation the 1-year mortality difference between women and men were no longer significant (adjusted OR:1.01, 95% CI:0.84–1.21, p = 0.929).

## Discussions

This registry showed a changing trend in sex differences in the baseline characteristics, risk factors, treatments and outcomes of ACS patients in Malaysia. Although some of the risk factors affected both sexes but they have different prevalence and impact.

### Baseline characteristics and risk factors

More men than women were enrolled in this registry, similar to many ACS registries [8, 24–26]. However, this over representation of men with ACS (men-to-women ratio of 4:1) was glaring when compared to the Malaysian population estimate for 2019 which has a men-to-women ratio of 107:100 [27]. Possible explanations for this low number of women enrolled include atypical presentation leading to misdiagnoses, failure to sought treatment due to ignorance or socio-economic status or cultural believes [28–30].

Like most other clinical trials and registries, women who present with ACS were older and have more comorbidities, including DM, hypertension, dyslipidemia, heart failure. Women were less likely to be smokers and less likely to have history of previous myocardial infarction [8, 9, 14, 28, 31]. In the STEMI cohort women now presented 8.28 years later than men cf. 7.30 years in the 2006–2010 cohort. This represent a slight improvement and is also similar to most western societies where ischemic heart disease develops on average 7–10 years later in women compared with men [12]. However, recently most likely due to unfavourable lifestyle changes over the past decade, manifestation of ischemic heart disease in younger women is increasing. Recent studies showed a significant increase in mortality rates of ACS in young women <55 years of age [4–6]. In our study although women < 50 years of age has higher mortality compared to men, however, the marked increase in mortality rate in women <40 years of age from the 2006–2010 cohort were not present anymore in the 2012–2016 cohort. The observed reduction in percentages of patients with underlying coronary risk factors like DM, hypertension, dyslipidemia, previous myocardial infarction and previous heart failure in part may reflect some success of primary and secondary preventive measures. These reductions may also indicate an increasing role being played by other risk factors which were not assessed in this study e.g. impaired glucose tolerance/pre-diabetes, non-traditional risk factors like higher levels of psychosocial burden for example depression or poor mental health. Although the latter increase the risk to develop coronary artery disease (CAD) to a similar degree in women and men but the prevalence is significantly higher in women, especially in younger women, and this leads to worse outcomes [32, 33].

The marked increase in the percentage of patients presenting with Killip class IV in the current cohort (2012–2016) of STEMI and NSTEMI patients may partly be due to lower prescriptions of evidence-based medicines (if the in-hospital prescription habits is any guide to go by) to control underlying coronary risk factors like DM, hypertension, prior myocardial infarction or prior heart failure, poorer pre-existing health as mentioned above and for women, additionally due to older age at presentation and multiple co- morbidities. Thus, the current cohort was comparatively sicker, thereby increasing their morbidity and mortality.

## In-hospital medications received

Treatment of ACS should follow current guidelines as medical treatment and intervention benefit both women and men [34–36]. It has been shown that thrombolytic therapy reduces mortality equally in both sexes [37]. Similar to other studies this registry showed that women were less likely to receive evidence-based medicines such as antiplatelets, beta-blocker, and ACE-I [38–40]. Compounding this factor is the further reduction in usage of these medicines for the 2012–2016 cohort. Possible explanation for this includes older age of patient [40]. Furthermore, this difference in prescription habit between periods could partly be explained by the inclusion of tertiary and secondary hospitals without cardiologist in the 2012–2016 cohort. Therefore, physician's awareness of evidence-based treatment, fear of adverse effects, and socioeconomic resources may contribute. These low usages of evidence-based medicine contributed to the higher unadjusted mortality rates and the adjusted mortality rates at 30-day and 1-year. On the contrary, more women were taking oral hypoglycaemic agents and insulin, diuretics and CCB reflecting their higher prevalence of DM, hypertension and heart failure.

## In-hospital procedures, culprit arteries and numbers of diseased vessels

It is well accepted that women derive the same benefits from PCI as men. Although there was an increase in the percentages of coronary angiography and percutaneous intervention being performed, women still received less invasive investigation and reperfusion procedures. The

findings of this registry are similar to most other studies [8, 38–42]. Possible explanations for these include: i) under-estimation of patient's risk due to misperception that CAD is a disease of men [9, 40, 43]; ii) atypical presentation of women with CAD complicating evaluation [7, 44]; iii) traditional screening tests for ischemia are less capable of detecting CAD in women [7, 37, 45]; iv) physician bias to refer or perform coronary angiography [43]; v) personal preference, women being less willing to undergo invasive procedures [46] and vii) patient's education, those with lower levels were less inclined to agree for cardiac catheterisation [47]. With the advancement in PCI technology and skills, cardiologists are performing more interventions. Insights from this registry will help us better understand our shortfalls and focus on proper and optimal utilisation of our resources.

Where previously (2006–2010 cohort) there were more patients with triple vessel disease, now there were more patients with single vessel disease. This current trend is similar to other studies involving western populations [14, 48]. Is this a price we have to pay for industrialisation and urbanisation, with the consequent change in lifestyle and socio-economic stressors?

## Treatment of STEMI

Women were more likely to be treated conservatively, with lower rates of primary PCI or no reperfusion therapy. These findings were similar to studies on ACS management performed in other developing countries [26, 49] and also elsewhere [41]. However, there were encouraging trends with increasing percentages of women receiving some form of reperfusion especially primary PCI. Although there were improvements in DTN and DTB times between the 2012–2016 cohort compared to the 2006–2010 cohort, the improvements were less for women and women still has longer DTN and DTB time.

These improvements were partly due to the initiation of the primary PCI referral network in May 2015 that was established in the Klang valley (central part of Malaysia) and ACS referral summits that were carried out throughout the country to raise awareness among healthcare workers on benefits of early referral and primary PCI during STEMI. With respect to women in particular, there were road shows that were carried out by the Women's Heart Health Organisation (WH$^2$O) at multiple sites throughout the country to healthcare workers and the lay public to raise awareness of the high incidence and mortality among women with coronary artery disease, in particular ACS. At the hospital level, PCI-capable centers like ours also establish clear cut pathway for management of STEMI patients, either direct walk-in or referral from district hospitals. The primary PCI referral networks were not rolled out for the whole country due to limitations in human resources and facilities.

## In-hospital clinical outcomes and mortality

Although there is overall benefit of invasive revascularisation, female sex has been associated with increased risk of bleeding and vascular complications [50, 51]. Also, it has been shown that women had a higher risk of bleeding with antithrombotic than men [52]. We had similar findings for women in the STEMI cohort and this could contribute to the higher mortality observed in women.

Overall, women had longer CCU and total hospital stay then men. This could be explained by their older age and more comorbidities at presentation. The shorter CCU stay for STEMI patients for both women and men could be explained by the increased percentage of patients undergoing primary PCI, thus shortening the recovery from the acute phase whereas, the longer total stay could partly be due to a higher percentage of patients presenting with Killip class IV. Whilst the longer CCU and total hospital stay for the NSTEMI and UA cohort could partly be due to a higher percentage of patients presenting with Killip class IV, and facility

limitations, as most PCI-capable centers in the country has only one to two cardiac catheterisation laboratories, thus, patients need to be kept longer in order for the case to be done within the index admissions.

There are conflicting evidence on effect of sex and gender on mortality following ACS. Generally, unadjusted mortality after ACS is higher for women compared to men [14, 28].

In this registry, women with STEMI had higher unadjusted in-hospital mortality rates across the 10 year span compared to men. After multiple logistic regression analyses, it can be clearly seen that in-hospital mortality was affected by age (women were older), clinical presentation at admission i.e. Killip classification, heart rate (HR), systolic blood pressure (SBP), elevated creatinine kinase (i.e. extend of myocardial damage) and pre-existing or underlying conditions like DM, hypertension, cigarette smoking and ethnicity. Of note, the unadjusted in-hospital mortality rates were higher for the STEMI cohort for both women and men in period 2 compared to period 1, despite the increase in rates of PCI and fibrinolysis. This could largely be explained by the markedly increased percentage of patients presenting with Killip class IV, in fact seen across the ACS stratum as mentioned above. After adjusting for these covariates, the difference was no longer significant. Findings of this registry on in-hospital mortality is in keeping with other registries which reported that women do not have worse outcome after acute MI when age and other factors are taken into account [8, 20, 28, 31]. This is in contrast to other studies involving some Western, Middle Eastern and Asian populations which found that women had higher in-hospital mortality even after adjusting for age and other covariates [9, 24, 41, 53].

By Day-30, in addition to the above factors, hospital management plays an important part i.e. the decision and swiftness to provide reperfusion therapy, prescription of guideline-directed medicines that reduce or has an impact on mortality and standard of care provided by institutions. Again, after adjusting for these the difference in mortality rate at Day-30 were no longer significant.

However, by one year, despite adjustments for all covariates including age, clinical presentation, coronary risk factors, hospital management and institutional factors, women still have significantly higher mortality rate. Thus, in addition to all the above-mentioned factors, there were factors beyond what we have adjusted for e.g. DTN and DTB times, bleeding complications, non-traditional or gender-specific factors that were responsible. The United States National Cardiovascular Data Registry demonstrated that from 2008–2014, contact-to-device time remained longer in women than men, and longer reperfusion time was associated with increased mortality for both women and men [54]. In another regional study using standardised PCI-based STEMI protocol, no sex differences in in-hospital or long-term (5-year) age-adjusted mortality were seen, suggesting mortality in women can be improved by using standardised STEMI protocols [55]. Also, sex differences in the pathophysiology of STEMI may lead to differences in response to existing guidelines for STEMI therapy. It was noted in some countries myocardial infarction with non-obstructive coronary arteries (MINOCA), which is associated with a 4.7% all-cause mortality at 1 year, is more common in women than men [56]. However, in our study, MINOCA is more common in women only in the NSTEMI cohort.

The 2006–2012 cohort showed a spike of increase in mortality in the <40-year-old women, similar to recent studies elsewhere that reported significant increase in mortality rates of acute coronary syndrome (ACS) in young women <55 years of age [4–6]. This phenomenon was also documented in another publication using this NCVD-ACS registry data. It was noted that prevalence of ACS in reproductive-age women were low, but their prognosis were worse than that of older women or same-aged men. It was postulated that this was probably due to the higher incidence of STEMI in this group [57]. In another study it was noted that nearly all patients aged 15–55 years with acute myocardial infarction had at least 1 cardiovascular risk

factor, but women were less likely than men to be told they were at risk or have a provider discussed risk modification with them [58]. From the same study, further analysis found that short- and long-term mortality rates were higher in young patients who exceeded the recommended reperfusion goals (particularly for the PCI transfer patients) compared with patients who met the recommended perfusion guidelines [59]. Women in our study had longer DTN and DTB time. This trend of marked increase in mortality in the younger-aged women was no longer present in the 2012–2016 cohort.

For the NSTEMI cohort, where previously (2006–2010) there was no significant difference in the in-hospital mortality between sexes, it was now (2012–2016) higher in women than men, although the difference was not big. This difference was in fact present through Day-30 and 1-year. However, after adjusting for age (women were older) and clinical presentation at admission, these differences were no longer significant. This change in trend is likely due to a higher percentage of women presenting with Killip Class IV in the current cohort and lower prescription of guideline-directed pharmacotherapy.

## Strengths

The NCVD-ACS registry enrols consecutive ACS patients admitted to multiple centers nationwide. Unlike randomized clinical trials that tend to exclude high-risk and elderly patients, this registry is an all-comers registry that collects data on the full spectrum of ACS patients in a real-world setting. Hence, this registry should reflect true sex and gender differences in the population. Findings of this registry should provide insights to healthcare workers and policy makers to improve standard and quality of care.

## Limitations

Women could have been under-represented in the NCVD-ACS registry as a result of selection bias due to educational, psychosocial or cultural factors. Despite our attempt to include as many hospitals as we could, participation is voluntary thus, there may be a selection bias, only larger hospital with better facilities and staffing will participate. Also, many private hospitals with significant numbers of ACS patients did not participate. Health care in private hospitals are mostly 'self-paying' (i.e. paid by individual or covered by personal or company insurance), whereas public hospitals or institutions were subsided either fully or partially by government funding. Thus, this registry may not reflect a wide range of patients with different socioeconomic status, education and occupations with possibly different cardiovascular risk profile and disease spectrum. Finally, errors in data entry cannot be completely ruled out and may result in unrecognized biases despite periodic audits.

## Conclusion

The higher mortality in women with ACS, in particular STEMI, has been attributed to a longer patient and system delay, older age, more clustering of cardiovascular risk factors, lower user of guideline-directed medical and invasive treatment, more bleeding complications and differences in STEMI pathophysiology. We need to take advantage of our knowledge of these sex and gender difference to improve outcome in women. These will include early risk factor identification and modification, optimal guideline-directed therapy, putting in place standardised PCI-based STEMI protocol (in PCI-capable facility) besides patient and physician education to narrow this sex and gender gap. Finally, inclusion of more women in research/clinical trials to understand better this disparity.

## Supporting information

**S1 Fig. Acute coronary syndrome by sex, 2012–2016.**
(TIF)

**S2 Fig. Proportion of ACS in-hospital mortality by age groups and sex, 2012–2016 (n = 35232).**
(TIF)

**S3 Fig. Acute coronary syndrome by sex comparing 2006–2010 and 2012–2016.**
(TIF)

**S1 Table. Treatment of STEMI, 2012–2016.**
(DOCX)

**S2 Table. Prognostic factors for in-hospital mortality by sex, 2012–2016.**
(DOCX)

## Acknowledgments

The authors would like to thank the Director General of Health Malaysia for the permission to publish this paper. The authors would also like to thank the principal investigators from all the participation centers for data collection: Omar Ismail, Cardiology Department, Hospital Pulau Pinang, George Town, Pulau Pinang; Sharifah Omar, Department of Medicine, Hospital Melaka, Melaka; Azhari Rosman, National Heart Institute, Kuala Lumpur, Wilayah Persekutuan Kuala Lumpur; Phanindtranath Mahadasa, Department of Medicine, Hospital Queen Elizabeth, Kota Kinabalu, Sabah; Monniaty Mohamed, Department of Medicine, Hospital Raja Perempuan Zainab II, Kota Bahru, Kelantan; K Chandran, Cardiology Unit, Department of Medicine, Hospital Raja Permaisuri Bainun, Ipoh, Perak; Sim Kui Hian, Department of Cardiology, Sarawak Heart Center, Kota Samarahan, Sarawak; Liew Houng Bang, Department of Cardiology, Hospital Queen Elizabeth II, Kota Kinabalu, Sabah; Abdul Kahar Abdul Ghapar, Department of Cardiology, Hospital Serdang, Kajang, Selangor; Hasmannizar Abd Manap, Cardiology Department, Hospital Sultanah Bahiyah, Kedah; Sapari Satwi, Department of Medicine, Hospital Tengku Ampuan Afzan, Kuantan, Pahang; Santha Kumari Natkunam, Department of Medicine, Hospital Tengku Ampuan Rahimah, Klang, Selangor; Sia Koon Ket, Department of Medicine, Hospital Tuanku Fauziah, Kangar, Perlis; K Sree Raman, Department of Medicine, Hospital Tuanku Ja'afar, Seremban, Negeri Sembilan; Oon Yen Yee, Department of Medicine, Hospital Ampang, Ampang Jaya, Selangor; Sazzli Shahlan Kasim, Cardiology Unit, Department of Medicine, UiTM Medical Center, Sungai Buloh, Selangor; Ng Wan Jun, Department of Medicine, Hospital Duchess of Kent, Sandakan, Sabah; Kew Chen Hui, Department of Medicine, Hospital Lahad Datu, Lahad Datu, Sabah; Muhamad Ali SK Abdul Kader, Department of Cardiology, Hospital Pulau Pinang, GeorgeTown, Pulau Pinang; Paras Doshi, Department of Medicine, Hospital Kuala Lumpur, Kuala Lumpur, Wilayah Persekutuan Kuala Lumpur; Ting Seng Kiat, Department of Medicine, Hospital Melaka, Melaka; Aizai Azan Abdul Rahim, National Heart Institute, Kuala Lumpur, Wilayah Persekutuan Kuala Lumpur; Ahmad Maujad Ali, Department of Medicine, Oriental Melaka Straits Medical Centre, Melaka; Michal Christina Steven, Department of Medicine, Hospital Queen Elizabeth, Kota Kinabalu, Sabah; Mansor Yahaya, Cardiology Unit, Department of Medicine, Hospital Raja Perempuan Zainab II, Kota Bahru, Kelantan; Asri Ranga Abdullah, Cardiology Unit, Department of Medicine, Hospital Raja Permaisuri Bainun; Ong Tiong Kiam, Department of Cardiology, Sarawak Heart Center, Kota Samarahan, Sarawak; Saravanan Krishinan, Department of Cardiology, Hospital Sultanah Bahiyah, Alor Setar, Kedah; Mohd Sapawi Mohamed,

Cardiology Unit, Department of Medicine, Hospital Sultanah Nur Zahirah, Kuala Trengganu, Trengganu; Siti Khairani Zainal Abidin, Department of Cardiology, Hospital Tengku Ampuan Afzan, Kuantan, Pahang; Oteh Maskon, Cardiology Unit, Department of Medicine, UKM Medical Center, Bangi, Selangor. Last but not least the authors would like to thank all physicians and nurses participating in the Malaysian NCVD-ACS registry and Ms. Gunavathy Selvaraj for coordinating data collection and data cleaning.

## Author Contributions

**Conceptualization:** Chuey Yan Lee.

**Data curation:** Kien Ting Liu.

**Formal analysis:** Kien Ting Liu.

**Investigation:** Chuey Yan Lee, Hou Tee Lu, Rosli Mohd Ali, Alan Yean Yip Fong, Wan Azman Wan Ahmad.

**Methodology:** Chuey Yan Lee, Hou Tee Lu.

**Project administration:** Chuey Yan Lee.

**Supervision:** Chuey Yan Lee.

**Visualization:** Chuey Yan Lee, Kien Ting Liu.

**Writing – original draft:** Chuey Yan Lee.

**Writing – review & editing:** Chuey Yan Lee, Hou Tee Lu, Rosli Mohd Ali, Alan Yean Yip Fong, Wan Azman Wan Ahmad.

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
