## [Decision Letter · Decision Letter 0]

31 Dec 2020

PONE-D-20-33875

Sex and gender differences in presentation, treatment and outcomes in acute coronary syndrome, a 10 year study from a multi-ethnic Asian population: the Malaysian National Cardiovascular Disease Database - Acute Coronary Syndrome (NCVD-ACS) Registry

PLOS ONE

Dear Dr. Lee,

Thank you for submitting your manuscript to PLOS ONE. After careful consideration, we feel that it has merit but does not fully meet PLOS ONE’s publication criteria as it currently stands. Therefore, we invite you to submit a revised version of the manuscript that addresses the points raised during the review process.

Overall the paper is well written and should be a nice addition to the literature. The authors should still address the contemporary nature of the cohort beyond 2010 and also statistical considerations for multiple comparisons. 

We look forward to receiving your revised manuscript.

Kind regards,

R. Jay Widmer

Academic Editor

PLOS ONE

Journal Requirements:

2. One of the noted authors is a group or consortium [NCVD Investigators]. In addition to naming the author group, please list the individual authors and affiliations within this group in the acknowledgments section of your manuscript. Please also indicate clearly a lead author for this group along with a contact email address.

Additional Editor Comments (if provided):

The authors are to be congradulated on such work. Overall the reviews were brief and positive. The authors should comment on multiple comparisons made and any statistical considerations taken. The authors should also consider a more contemporary cohort beyond 2010.

Reviewers' comments:

Reviewer's Responses to Questions

**Comments to the Author**

1. Is the manuscript technically sound, and do the data support the conclusions?

Reviewer #1: Yes

Reviewer #2: Yes

2. Has the statistical analysis been performed appropriately and rigorously? 

Reviewer #1: Yes

Reviewer #2: Yes

3. Have the authors made all data underlying the findings in their manuscript fully available?

Reviewer #1: Yes

Reviewer #2: Yes

4. Is the manuscript presented in an intelligible fashion and written in standard English?

Reviewer #1: Yes

Reviewer #2: Yes

5. Review Comments to the Author

Reviewer #1: Good original article on multi-ethic Asian demographic. Large sample size over a good period with long term outcome comparison. Clear illustration between differences in gender management, and outcomes. Clear comparison with Western data.

Reviewer #2: Insightful on local data collected.

Good comparison on over 5 years period - (2006-2010) and (2012-2016). However, how about the continuation of data from March 2010 to Dec 2011?

Overall well presented with a lot of information.

6. PLOS authors have the option to publish the peer review history of their article (what does this mean?). If published, this will include your full peer review and any attached files.

Reviewer #1: **Yes: **Nicholas Yul Chye Chua

Reviewer #2: **Yes: **WAI SUN CHOO

---

## [Author Response · Author response to Decision Letter 0]

17 Jan 2021

Dear Editor and reviewers,

Thank you for your kind comments. Following are my response to the queries raised:

1) During the writing of the first manuscript, Sex differences in acute coronary syndrome in a multi-ethnic Asian population: Results of the Malaysian National Cardiovascular Disease Database-Acute Coronary Syndrome (NCVD-ACS) Registry, February 2010 was the latest available clean data. Likewise, during the writing of the present manuscript December 2016 was the latest available clean data. As we wanted to compare two equivalent time period, we therefore choose 2012 to 2016. The data from March 2010 to Dec 2011 are available in the Annual Report of the NCVD-ACS Registry published and available online at http://www.acrm.org.my/ncvd.

2) Statistical analyses comparing women and men on the various parameters were done for the first cohort from 2006 to 2010 and second cohort from 2012 to 2016 as described in the methods section. As my aim was to compare these two periods to see if there is any change in trends in the patients’ characteristics, clinical presentations, treatments and outcomes, no further statistical analysis were done. To enable readers to see the trend easily, we provide multiple bar charts with the actual value of the variables shown at the bottom of the chart for the more avid readers who prefer to read numbers. 

3) Additional requirements:

i) I have checked to ensure the manuscript meets PLOS ONE’s style requirements.

ii) Regarding the authors list, I am the first author and the other 5 co-authors are as listed. There is no group involved in the writing of this manuscript. I wanted to acknowledge the principal investigators from each of the centers that contributed raw data to the NCVD-ACS registry but did not fulfil authorship criteria. Since they do not meet authorship requirement I will mention their names in the acknowledgement section as suggested. My apologies for the mistake.

Thank you. 

Sincerely,

Dr Chuey Yan Lee.

---

## [Editor Report · Decision Letter 1]

20 Jan 2021

Sex and gender differences in presentation, treatment and outcomes in acute coronary syndrome, a 10 year study from a multi-ethnic Asian population: the Malaysian National Cardiovascular Disease Database - Acute Coronary Syndrome (NCVD-ACS) Registry

PONE-D-20-33875R1

Dear Dr. Lee,

We’re pleased to inform you that your manuscript has been judged scientifically suitable for publication and will be formally accepted for publication once it meets all outstanding technical requirements.

Kind regards,

R. Jay Widmer

Academic Editor

PLOS ONE

---

## [Editor Report · Acceptance letter]

28 Jan 2021

PONE-D-20-33875R1 

Sex and gender differences in presentation, treatment and outcomes in acute coronary syndrome, a 10 year study from a multi-ethnic Asian population: the Malaysian National Cardiovascular Disease Database - Acute Coronary Syndrome (NCVD-ACS) Registry 

Dear Dr. Lee:

I'm pleased to inform you that your manuscript has been deemed suitable for publication in PLOS ONE. Congratulations! Your manuscript is now with our production department. 

Kind regards, 

on behalf of

Dr. R. Jay Widmer 

Academic Editor

PLOS ONE